# Failure-Mode Shift of Metal/Composite L-Joint with Grooved Structure under Compressive Load

**DOI:** 10.3390/polym14051051

**Published:** 2022-03-06

**Authors:** Zhenhang Kang, Zhu Liu, Yongpeng Lei, Jifeng Zhang

**Affiliations:** Key Laboratory of Advanced Ship Materials and Mechanics, College of Aerospace and Civil Engineering, Harbin Engineering University, Harbin 150001, China; kangzhenhang@hrbeu.edu.cn (Z.K.); liuzhu0618@hrbeu.edu.cn (Z.L.); yongpenglei@126.com (Y.L.)

**Keywords:** metal/composite, L-joint, bonding length, failure mode, equivalent, RVE

## Abstract

Bond length and bond interface morphology have a great influence on the performance of metal/composite hybrid joints. In this paper, a metal/composite L-joint with groove structure was designed, and seven groups with different bonding lengths were fabricated using the VARI (Vacuum Assisted Resin Infusion) process to study the effect of different bonding lengths on the performance of the joint. In the simulation analysis of the metal/composite L-joint, the stiffness equivalence method was adopted, and the groove structure was equivalent to a 0-thickness element layer. The applicability of the simulation method was verified by comparing the ultimate load, displacement and failure mode of the test and simulation. Furthermore, the simulation method was used to simulate more compression experiments of metal/composite L-joints with different bonding lengths, and prediction diagrams of failure displacement and failure mode were produced. According to the prediction map, when the bonding length is 100.00 mm, the metal/composite L-joint has better compressive properties.

## 1. Introduction

Fiber-reinforced composite materials are widely used in modern ships for their outstanding performance in strength/weight ratio, design flexibility and electromagnetic resistance [1,2,3]. Therefore, the connection between metals and composite materials is an inevitable problem. In marine structures, L-shaped joints and similar components are very common, for example, they are easy to find at the intersection between the board edge and the upper deck [4,5]. The design method and performance evaluation of similar steel structures are simple, mature and accurate for engineering. However, for metal/composite structures, due to their many components and complex damage modes, they mostly rely on subjective experience or direct model tests.

As one of the basic structural components of large-scale composite marine structures, the mechanics of L-joints have been studied more in [6,7,8]. Feih S. and H.R. Shercliff [9,10] predicted the failure behavior of composite L-joint structures under tensile load. The results show that the zone criterion can predict the failure load of joints with different fillet shapes very well, and the damage mechanics method can be used to simulate the crack propagation prediction during failure of the adhesive. In addition, they used ABAQUS to predict the damage and failure of the L-joint. They used a numerical model to evaluate the effects of parameters such as curvature, thickness, stacking order and combined load on the failure mode of the L-joints. Qin Kai and Yan Renjun [11] studied the transition of the failure mode of the sandwich-composite L-shaped joint of the hull structure under tensile load. They manufactured L-joints with transition zone radii (R) of 45 mm, 90 mm and 180 mm, respectively; they then analyzed and summarized the failure modes of the L-joints. Finally, through numerical simulation, the failure mode of L-joints with a radius from 45 mm to 180 mm was predicted. Haiyan Zeng and Renjun Yan [12] studied the failure prediction of sandwich-composite L-shaped joints under bending and established a numerical model to predict the bearing capacity of the joints. The feasibility of the proposed empirical formula was verified through experiments. Many scholars have studied the superstructure of ship structures, focusing on the mechanical behavior of sandwich-composite L-joints. The connection behavior of the composite structure and the hull is mainly a bolted connection and a hybrid connection. In the current paper, a metal/composite bonded L-joint with groove structure is proposed, and its failure behavior is studied. This study may provide a reference for the connection between superstructure and ship structure

The purpose of this paper is to study the compressive damage behavior of metal/composite L-joints and further investigate the effect of bond length for possible design recommendations. In the study, a metal/composite L-joint structure design with groove structure was proposed, and the L-joint was fabricated using the VARI co-curing process. To study the effect of bond length on the compressive properties of metal/composite L-joints, seven bond joints with different lengths were designed, and their failure modes were analyzed in detail. In the simulation analysis of metal/composite L-joints, due to the complexity of the groove structure and the multi-faceted and multi-scale problems, the parameter study of the groove structure required lots of calculations. Therefore, the stiffness equivalence method was adopted, and the groove structure was equivalent to a 0-thickness cohesive layer to facilitate numerical analysis. The parameters of the 0-thickness cohesive element layer were calculated using the RVE method. The feasibility of this method was verified by comparison between the experiment and the simulation.

## 2. Specimen Manufacturing and Testing

The metal/composite L-joint consisted of a metal corner and two composite laminates. The L- type metal crutch was a right-angled connector, and its size and schematic diagram are shown in Figure 1 (bonding length = 55.00 mm). As shown in Figure 1, the thickness of the corners of the metal parts was 8.00 mm. The bonding area of the metal component was composed of four identical groove structures. The thickness of the groove structure was 3.00 mm, and the distance between adjacent ones was 2.00 mm.

The ±45° groove structure was designed in the L-shaped metal component, as shown in Figure 2. The value of 45° was relative to the boundary of the groove structure. Both sides of the bonding component had groove structure, which can eliminate the warpage caused by the groove structure. The size of the groove was determined based on previous research. The depth and width of the ±45° groove were 0.75 mm and 1.414 mm, respectively, and the cross-sectional shape of the groove was rectangular. The distance between adjacent grooves was 8.00 mm.

In addition, to explore the influence of the bonding length on the L-shaped joint, bonding components of other lengths were also designed, as shown in Figure 3. A total of seven metal components with different bonding lengths were designed, and the bonding lengths were 25.00 mm, 40.00 mm, 55.00 mm, 70.00 mm, 85.00 mm, 100.00 mm, and 115.00 mm, respectively.

The metal/composite L-joint was made using a vacuum-assisted resin infusion (VARI) process, and there were three specimens for each type of L-joint. The schematic diagram of the finished product is shown in Figure 4. The composite component was composed of two layers of 3.00 mm and three layers of 2.00 mm composite laminates. The composite material was glass-fiber-reinforced plastic (GFRP) (MingRen composite, Harbin, China). The fiber fabric, with a repeating length unit of T = 7.8 mm, was a plain bi-directional woven fabric glass fiber, with a thickness of t = 0.15 mm and a surface density of 600 g/m^2^ [13,14]. The performance parameters of the glass fiber reinforced plastic layer were $E_{11}$ = 20 GPa, $E_{22}$ = $E_{33}$ = 6.545 GPa, $G_{12}$ = $G_{23}$ = 3.545 GPa, $G_{23}$ = 1.52 GPa, $\nu_{12}$ = $\nu_{13}$ = 0.3, $\nu_{23}$ = 0.45. In addition, the resin matrix was made of Ashland Derakane™ 411 epoxy vinyl ester resin(Ashland, Catlettsburg, KY, USA), which is used to manufacture glass-fiber-reinforced composite laminates, and the material parameters of the resin cast body were E = 2.9 GPa and λ = 0.3. The adhesive layer structure was formed during the curing process of the vinyl resin.

To verify the ultimate compressive strength of the L-joint, a compression experiment was designed. Experiments on the L-joints were conducted on the Instron 8001 universal testing machine (Instron, Norwood, MA, USA) shown in Figure 5. During the experiments, the two ends of the L-joint were clamped by premade steel fixtures. The lower end was latched onto the test bench, and the upper end was latched onto the actuator. The applied load was controlled through the displacement of the actuator, which was set at a speed of 2.5 mm/min.

## 3. Simulation Study

### 3.1. Establishment of RVE Model

The representative volume element (RVE) of the bonded part was intercepted in the study, as shown in Figure 6. In Figure 6, the established RVE model is a rectangular parallelepiped of 10.00 mm × 10.00 mm × 3.00 mm. The ±45° groove structure was symmetrical diagonally; the depth and width of the groove were 0.75 mm and 1.414 mm, respectively; and the thickness of the adhesive layer was 0.75 mm.

When using RVE for structural analysis, periodic boundary conditions are usually applied to the elements to achieve deformation control of the numerical model. Periodic boundary conditions can coordinate the deformation of the adjacent contact surfaces of RVE so that the internal deformation of the entire material model has continuity.

The symmetry boundary surface of the RVE must have the same mesh, to ensure that any node on the boundary surface can find its corresponding node on its symmetry surface. Assuming that $P_{1}$ is a node on a certain boundary surface of the RVE and $P_{2}$ is the node corresponding to it on the symmetry plane, the positions of the node $P_{1}$ and $P_{2}$ must satisfy the relationship, as follows in Equation (1):(1)$$\left\{ \begin{array}{l} x_{i}^{P_{1}}=x_{i}^{P_{2}}\\ x_{j}^{P_{1}}=x_{j}^{P_{2}}\\ x_{k}^{P_{1}}=x_{k}^{P_{2}}\pm 1 \end{array}\right.\;\;\;\left(i,j,k=1,2,3\mathrm{and}\;i\neq j\neq k\right)$$
where $x_{i}^{P_{1}}$,$x_{j}^{P_{1}}$,$x_{k}^{P_{1}}$ and $x_{i}^{P_{2}}$,$x_{j}^{P_{2}}$,$x_{k}^{P_{2}}$ are the node coordinates of $P_{1}$ and $P_{2}$, respectively. Assuming finite deformation and applying arbitrary average deformation gradient tensor $\bar{F}$ to RVE, the periodic boundary condition can be expressed as Equation (2):(2)$$\begin{array}{l} x\left(P_{1}\right)-x\left(P_{2}\right)=\bar{F}\left[X\left(P_{1}\right)-X\left(P_{2}\right)\right]\\ P\left(P_{1}\right)=-P\left(P_{2}\right) \end{array}$$
where *X* and x represent the position vectors of the nodes on the boundary surface before and after deformation, respectively, and P represent the load acting on the corresponding nodes.

In Abaqus, the application of periodic boundary conditions of representative volume elements is realized by introducing reference points and establishing parameter constraints. To express the form and application process of periodic boundary conditions more clearly, a representative volume element was established, as illustrated in Figure 7.

$X_{1},X_{2},Y_{1},Y_{2},Z_{1}$ and $Z_{2}$ are used to represent the node sets of the front and rear boundary surfaces, left and right interfaces, and upper and lower boundary surfaces of the REV, respectively. $R_{1},R_{2}$ and $R_{3}$ are used to represent the reference points associated with the corresponding boundary surfaces. Then the constraint equation of the periodic boundary condition can be expressed as Equation (3):(3)$$\left\{ \begin{array}{l} u_{1}^{X_{1}}-u_{1}^{X_{2}}=u_{1}^{R_{1}}\\ u_{2}^{X_{1}}-u_{2}^{X_{2}}=u_{2}^{R_{1}}\\ u_{3}^{X_{1}}-u_{3}^{X_{2}}=u_{3}^{R_{1}} \end{array}\right.,\left\{ \begin{array}{l} u_{1}^{Y_{1}}-u_{1}^{Y_{2}}=u_{1}^{R_{2}}\\ u_{2}^{Y_{1}}-u_{2}^{Y_{2}}=u_{2}^{R_{2}}\\ u_{3}^{Y_{1}}-u_{3}^{Y_{2}}=u_{3}^{R_{2}} \end{array}\right.,\left\{ \begin{array}{l} u_{1}^{Z_{1}}-u_{1}^{Z_{2}}=u_{1}^{R_{3}}\\ u_{2}^{Z_{1}}-u_{2}^{Z_{2}}=u_{2}^{R_{3}}\\ u_{3}^{Z_{1}}-u_{3}^{Z_{2}}=u_{3}^{R_{3}} \end{array}\right.$$

The established RVE numerical model consisted of three parts: the composite component, the adhesive layer structure and the metal component. The whole was a rectangular parallelepiped of 10.00 mm × 10.00 mm × 3.00 mm. These parts consisted of 394,224 C3D8R elements (as shown in Figure 7). In addition, a 0-thickness cohesive layer was defined between the metal component and the adhesive layer structure (red area in Figure 7); a tie constraint was applied between the composite component and the glue layer structure, enabling hard contact between components.

### 3.2. Material Parameters of the Equivalent Cohesive Layer

To facilitate the analysis of complex contact surfaces, the complex groove structure was made equivalent to a 0-thickness cohesive element layer by a direct equivalent method. The direct equivalent method was used to calculate the equivalent stress and strain directly, based on the average surface or volume of the field, such as stress and strain. It was then used to solve the macro equivalent performance based on the relationship between the macro equivalent stress and the macro equivalent strain [15].

For simple structures, the material mechanics method or elastic mechanics method can be used to directly solve the stress and strain of each component, then average the macro equivalent stress $\bar{\sigma }$; finally, the equivalent stiffness coefficient $\bar{C}$ of the composite material can be solved according to the definition, as shown in Equation (4):(4)$$\bar{\sigma }=\bar{C}:\bar{\varepsilon }$$
where $\bar{\sigma }$ and $\bar{\varepsilon }$ represent the equivalent macroscopic stress and equivalent strain, respectively, and $\bar{C}$ represents the equivalent macroscopic stiffness tensor. The direct equivalent method is mainly suitable for solving the equivalent stiffness of composite materials with simple structures. Therefore, in practical applications, the direct equivalent method is mainly used for the simple estimation of equivalent stiffness.

To simulate the damage failure of laminated composites in the tensile shear test, the cohesive zone model is considered. In addition to stiffness, the initiation criteria and energy-based propagation criteria for crack initiation are defined. The criterion for the initiation of interlayer damage is the law of traction separation, as shown in Equation (5):(5)$${\left(\frac{\sigma_{n}}{N}\right)}^{2}+{\left(\frac{\sigma_{t}}{T}\right)}^{2}+{\left(\frac{\sigma_{s}}{S}\right)}^{2}=1$$
where $\sigma_{n}$,$\sigma_{t}$, are $\sigma_{s}$ are, respectively, the tensile stress relative to the normal direction n and shear directions t and s. N, T and S stand for their critical values. The law of exponential damage evolution based on the Power Low energy criterion (Equation (6)), linear softening and mixed-mode is selected.
(6)$$\frac{G_{\varepsilon }}{G_{\varepsilon C}}={\left(\frac{G_{I}}{G_{IC}}\right)}^{a_{m}}+{\left(\frac{G_{II}}{G_{IIC}}\right)}^{a_{n}}+{\left(\frac{G_{III}}{G_{IIIC}}\right)}^{a_{o}}$$
where $G_{\varepsilon }=G_{I}+G_{II}+G_{III}$ refers to the total energy release rate, and the value of exponent $a_{m}$, $a_{n}$ and $a_{o}$ are usually selected to be either 1 or 2. The areas under the traction-relative displacement curves for modes I, II or III are the relative critical energies released at failure $G_{IC}$, $G_{IIC}$ and $G_{IIIC}$, as shown in Equations (7) and (8):(7)$$G_{IC}=\frac{N\delta_{I}^{F}}{2}$$
(8)$$G_{IIC}=G_{IIIC}=\frac{T\delta_{II}^{F}}{2}$$
where $\delta_{I}^{F}$ and $\delta_{II}^{F}$ are the ultimate opening and tangential displacements, respectively [16]. When the value of the degradation criterion parameter $SDEG=\frac{G_{\varepsilon }}{G_{\varepsilon C}}=1$, the 0-thickness cohesive layer completely fails.

In this study, shear and pull-out simulations of the ±45° groove structure was carried out, and the load–displacement curve obtained is shown in Figure 8. According to the obtained data and Equations (7) and (8) above, the equivalent zero-layer viscous unit layer parameters are calculated, as shown in Table 1.

### 3.3. Simulation of Equivalent Metal/GFRP Joints

In this paper, the ABAQUS finite element software was used to numerically analyze the compression experiment of the metal-composite L-joint, and an equal-scale numerical model was established. The loading condition was the same as the experimental condition, which was 2.50 mm/min, as shown in Figure 9. In Figure 9, the boundary conditions of the fixed end were U1 = U2 = U3 = 0, UR1 = UR2 = 0. The boundary conditions of the compressive load loading end were U3 = 0, UR1 = UR2 = 0. The metal-composite L-joint model consisted of metal parts (Figure 9a), GFRP parts (Figure 9b,c) and interfacial cohesive element layers (Figure 9d–f). Metal parts and GFRP parts were divided into 24,675 meshes with element type C3D8R. The metal part (Figure 9a) is composed of two metal groove plates with a thickness of 3.00 mm and a 90° corner with a thickness of 8.00 mm. The failure of the metal structure adopted the ductile damage criterion, and the criterion for the onset of damage, the equivalent plastic strain (${\bar{\varepsilon }}_{D}^{pl}$), η, was a function of the stress triaxiality, η=−p/q. Stress triaxiality is a parameter that describes the triaxial stress state of a material, such as in pure shear, uniaxial tension (compression), equi-biaxial tension (compression), and equi-triaxial tension (compression) stress states. The values of η were 0, $\pm \frac{1}{3}$, $\pm \frac{2}{3}$ and ±∞, respectively. The GFRP parts consisted of three layers of 2.00 mm composite board (Figure 9c) and two layers of 3.00 mm composite board (Figure 9b). To predict the progressive damage of GFRP, the 3DHashin failure [17,18] criterion model (used by Puck for matrix failure) was considered. In the process of loading, when some damage was produced, the material properties were adjusted by the corresponding material performance degradation criteria to realize the failure procedure [19,20]. The continuous damage-mechanics constitutive model of GFRP structure was realized by the VUMAT (Vectorized user-material) subroutine in Abaqus. The expression of the 3D Hashin failure criterion is shown in Equations (9)–(14), and the degradation criterion is shown in Table 2.

Fiber tensile failure ($\sigma_{11}⩾0$):(9)$${\left(\frac{\sigma_{11}}{X_{t}}\right)}^{2}+{\left(\frac{\tau_{12}}{S_{12}}\right)}^{2}+{\left(\frac{\tau_{13}}{S_{13}}\right)}^{2}=1$$

Fiber compression failure ($\sigma_{11}<0$):(10)$$-\left(\frac{\sigma_{11}}{X_{c}}\right)=1$$

Matrix tensile failure ($\sigma_{22}+\sigma_{33}⩾0$):(11)$${\left(\frac{\sigma_{22}+\sigma_{33}}{Y_{t}}\right)}^{2}+\left(\frac{1}{S_{23}^{2}}\right)\left(\tau_{23}^{2}-\sigma_{22}\sigma_{33}\right)+{\left(\frac{\tau_{12}}{S_{12}}\right)}^{2}+{\left(\frac{\tau_{13}}{S_{13}}\right)}^{2}=1$$

Matrix compression failure ($\sigma_{22}+\sigma_{33}<0$):(12)$$\frac{1}{Y_{c}}\left[{\left(\frac{Y_{c}}{2S_{c}}\right)}^{2}-1\right](\sigma_{22}+\sigma_{33})+{\left(\frac{\sigma_{22}+\sigma_{33}}{2S_{12}}\right)}^{2}+\left(\frac{1}{S_{23}^{2}}\right)(\tau_{23}^{2}-\sigma_{22}\sigma_{33})+{\left(\frac{\tau_{12}}{S_{12}}\right)}^{2}+{\left(\frac{\tau_{13}}{S_{13}}\right)}^{2}=1$$

Tensile delamination failure ($\sigma_{33}⩾0$):(13)$${\left(\frac{\sigma_{33}}{Z_{t}}\right)}^{2}+{\left(\frac{\tau_{12}}{S_{13}}\right)}^{2}+{\left(\frac{\tau_{13}}{S_{23}}\right)}^{2}=1$$

Compression delamination failure ($\sigma_{33}<0$):(14)$${\left(\frac{\tau_{12}}{S_{13}}\right)}^{2}+{\left(\frac{\tau_{13}}{S_{23}}\right)}^{2}=1$$
where, $\sigma_{11}$,$\sigma_{22}$,$\sigma_{33}$,$\tau_{12}$,$\tau_{13}$ and $\tau_{23}$ are the principal direction stress of the composite, 1 represents the fiber direction, 2 represents the direction perpendicular to the fiber, and 3 represents perpendicularity to the 1 and 2 planes; $X_{t}$, $Y_{t}$ and $Z_{t}$ are the tensile strength in the main direction of the composite; $X_{c}$, $Y_{c}$ and $Z_{c}$ are the compressive strengths in the principal direction of the composite; and $S_{13}$, $S_{23}$ and $S_{12}$ are the shear strengths of the composite.

The interfacial cohesive element layer structure was mainly composed of three different cohesive layers, namely: the interfacial cohesive layer between the metal plane and GFRP (Figure 9d), the equivalent 0-thickness cohesive element layer (Figure 9e), and the interfacial cohesive elemental layer of GFRP (Figure 9f). The purpose of the interfacial cohesive layer between the metal plane and GFRP was to simulate the viscous behavior between the untreated metal surface and GFRP (Figure 9d); and the equivalent 0-thickness cohesive element layer was to simulate the fracture of the cured adhesive layer resin in the groove structure (Figure 9e); and the interfacial cohesive element layer of GFRP was to simulate the delamination failure of GFRP (Figure 9f). The COH3D8 element format was adopted for the cohesive elements. The bilinear shape law was generally used owing to the analysis time and convergence problem [21,22,23].

## 4. Results and Discussion 

To facilitate the elaboration and analysis of the failure mode of the metal-composite L-joints, the structure of the L-joint is disassembled and named, respectively, as shown in Figure 10. The composition of the metallic and GFRP components is marked in Figure 10, and the Cohesive structure consists of three different Cohesive elements (refer to Figure 9), defined as C-1, C-2, and C-3. Taking C-1 as an example, it consists of four independent layers of Cohesive elements, defined as C-1-1, C-1-2, C-1-3 and C-1-4, respectively.

### 4.1. Analysis of Experimental and Simulation Results of L-Joints with Different Bonding Lengths

When the L-joints were damaged and failed, neither the metal structure nor the GFRP structure was significantly damaged, so the failure mode analysis was based on the equivalent 0-thickness cohesion element layer. According to different failure modes, L-joints with different bonding lengths are divided into three groups for discussion.

#### 4.1.1. L-Joint with Bonding Length of 25.00 mm 

Figure 11 shows the load–displacement curve for an L-joint with a bond length of 25.00 mm. In Figure 11, the simulated and experimental ultimate loads are approximately 1652.02 N and 1613.74 N, a difference of 2.37%. However, the overall stiffness in the finite element model is higher than in the experiment, and there are two main reasons for the stiffness error. First, due to the unique geometry of the L-joint, after the initial compressive load was applied, small defects were created in the bond area, which can cause a significant drop in stiffness. Second, since the in-plane shear was nonlinear, it mainly existed in the complex ±45° groove structure of the bonding interface, and the nonlinearity became more significant as the load increased. As a combined effect, the displacement results in the experiment are larger than the simulation results. The displacements when the simulation and experiment reach the ultimate load are 4.83 mm and 5.34 mm, respectively, with a difference of 9.6%.

In Figure 11, the first load dump is due to the complete failure of C-1-1 and C-1-2, resulting in complete separation of the adhesive interface between Metal (3.00 mm)-1, GFRP (2.00 mm)-1 and GFRP (2.00 mm)-2. Obvious separation phenomena can be observed on both the experimental and simulated specimens. With the increase in compressive displacement, C-1-4 partially failed, resulting in incomplete separation of the bonding interface of Metal (3.00 mm)-2 and GFRP (2.00 mm)-3, thus forming the second load mutation. The third load dump was due to the complete failure of C-1-3 and the complete separation of the bonding interface of Metal (3.00 mm)-2 and GFRP (2.00 mm)-2. Obvious cracks were observed on both the experimental and simulated specimens. In the final stage of the experiment and simulation, C-1-4 was still under shear load, and C-3-1 was under peel load, resulting in the plasticity of Metal (3.00 mm)-1 and Metal (3.00 mm)-2; however, the displacement increased and the compressive load remained stable. The simulations agree with the experimental failure modes.

Figure 12 shows the load–displacement curve and its first-order differential curve of the L-joint compression simulation. Observing Figure 11, no matter the experimental or simulated compressive load, there is an obvious plateau before the sudden change. Therefore, the load–displacement curve of the compressive simulation is derived, and its first-order differential curve is obtained. A, B, and C in Figure 12, respectively, frame the stable load region. On the first-order differential curve, at their corresponding displacements, their derivatives are significantly reduced and greater than 0. We analyzed the entire process of the simulation and found that a small plastic deformation occurred in Metal (3.00 mm) before C-1 failed, and we believe that this was the cause of this phenomenon.

#### 4.1.2. L-Joint with Bonding Length of 40.00 mm, 55.00 mm and 70.00 mm 

Table 3 shows the comparison of the ultimate loads and corresponding displacements of different L-joint experiments and simulation curves. In Table 3, the deviation ratio between the experimental and simulated ultimate loads and their compressive displacements is within an acceptable range, indicating that the simulation can reflect the experimental results to a certain extent.

Figure 13 shows load–displacement curves for L-joints with different bonding lengths. In Figure 13, there is a relatively obvious slow growth phase before the first load dump, and the reason for the dump is more complicated than that of the L-joint with a bond length of 25 mm. When the load is abruptly changed, C-1-1, C-1-2 and C-1-3 all partially failed, and the SDEG value of C-1-1 was lower, although the compressive load decreased significantly; however, there was no complete separation of the bonded interface. In Figure 13, C-1-1 has less damage failure in the first load change, and peeling can be observed at the bonding edge of GFRP (2.00 mm)-2 and Metal (3.00 mm)-1. As the failures of C-1-2 and C-1-3 were extended, a second load mutation occurred, at which time, the bonding interface of GFRP (2.00 mm)-2 and Metal (3.00 mm) was completely separated; this can be observed in both experiments and simulations on obvious cracks. When the curve stabilizes, the L-joint enters a stable failure phase. Overall, the experimental and simulated failure modes are about the same.

Figure 14 shows the load–displacement curve and its first derivative curve of the compression simulation of the L-joint with different bonding lengths. Observing Figure 13, the loads of the experimental and simulated curves have a relatively gentle change stage, and we analyze the failure mode of the L-joint. In Figure 14, the load changes from A to B, and C-1-1, C-1-2 and C-1-3 both fail; furthermore, the plastic deformation of the metal part is also transferred from the two sides of the Metal (8.00 mm) to the outside and the inside of the Metal (8.00 mm), and finally transferred to the connection area with the metal (3.00 mm). This shows that when C-1-1 does not fail, Metal (8.00 mm) will bear the compressive load; when the load suddenly changes, C-1-1 gradually fails until it is completely deleted, and the plastic deformation of metal parts is also transferred from Metal (8.00 mm) to its connection area. From this, it can be inferred that the stress form of C-1-1 changes from interfacial peeling to interfacial shearing due to the change in the bonding length. Since the interfacial shear strength of C-1 is greater than the interfacial peel strength, C-1-1 is not easy to damage or destroy; conversely, C-1-2 is still in a state of interfacial peeling, so it is easier for it to fail.

#### 4.1.3. L-Joint with Bonding Length of 85.00 mm, 100.00 mm and 115.00 mm 

Table 4 shows the comparison of the ultimate loads and corresponding displacements of different L-joint experiments and simulation curves. In Table 4, the difference rate is also suitable, and the ultimate load and compression displacements are larger than those in Table 3, indicating that the mechanical properties of metal-composite L-joints improve with an increase in bonding length.

Figure 15 shows load–displacement curves for L-joints with different bonding lengths. In Figure 15a,b, the load variations of L-joints were mainly caused by the failure of C-1-2 and C-1-3. From the initial partial failure of C-1-1, C-1-2 and C-1-3 to the complete failure of C-1-2 and C-1-3, the damage of the overall structure occurred from partial debonding of GFRP (2.00 mm)-1 with Metal (3.00 mm)-1 to complete failure of GFRP (2.00 mm)-2 with Metal (3.00 mm). Clear cracks can be seen in Figure 15a,b. In Figure 15, partial failure of C-1-1 also occurred in the initial stage, and a clear small-scale peeling phenomenon was found in both experiments and simulations. The incomplete failure of C-1-1 caused the continuous bearing of Metal (8.00 mm); thus, the final stage of the load–displacement curve no longer fluctuates wildly. When the experiments and simulations were over, the metal-composite L-joint had higher residual strength and larger compressive displacement, which were maintained for a longer time.

Figure 16 shows the load–displacement curve and its first derivative curve of the compression simulation of the L-joint with different bonding lengths. In Figure 16a,b, since C-1-1 did not fail completely, Metal (8.00 mm) was continuously subjected to bending loads and its plastic deformation continued to increase. With the failure of C-1-2 and C-1-3, the plastic deformation at the junction of Metal (3.00 mm) and Metal (8.00 mm) also increased. In Figure 16, the plastic deformation of the metal part continues until the end of the experiment and simulation. This means that in the experiments and simulations, the metal parts undergo severe bending deformation, and the L-joint specimens undergo significant overall inward bending. Therefore, although the metal-composite L-joint did not experience a sudden change in load, we stopped the experiments and simulations and considered the L-joint to have failed.

### 4.2. Relationship between Failure Mode and Compressive Load for L-Joints

Table 5 shows the different structural failure conditions in the joint, and its corresponding compressive load before the L-joint reaches the ultimate load. In Table 5, when the bonding length is 40.00–100.00 mm, both C-1-1, C-1-2 and C-1-3 have initial failure; when the bonding length is greater than 85.00 mm, the Metal (8.00 mm) has initial failure. The Metal (3.00 mm) did not fail until all L-joints reached their ultimate load.

The damage and failure processes of metal-composite L-joints with different bond lengths are still questionable. When the bonding length is from 25.00 mm to 115.00 mm, there must be a critical point, which can determine the bonding length when different failure modes are converted. To find the critical point, more bond lengths were designed in the paper; the compression simulation of the L-joint was performed, and the corresponding loads of damage initiation and final failure were recorded, as shown in Figure 17. Figure 17 shows a damage and failure prediction diagram for the metal-composite L-joints. In Figure 17, when the bond length is 40.00 mm and 95.00 mm, the ultimate load of the L-joint changes greatly, and the main failure mode also shifts accordingly. When the bonding length is from 40.00 to 80.00 mm, the initial failure of C-1-3 occurs, and the ultimate load of the L-joint increases and remains stable. When the bond length is from 85.00 to 120.00 mm, the Metal (8.00 mm) has an initial failure; the ultimate load further increases, and finally reaches the maximum at the bond length of 100.00 mm, then remains stable.

### 4.3. Relationship between Failure Mode and Compressive Displacement for L-Joints

Combined with the analysis in Section 3.1, it is concluded that different failure modes are closely related to compression displacement, as shown in Table 6. In Table 6, light green indicates partial failure, dark green indicates larger failure, and yellow indicates complete failure. When the metal-composite L-joint is subjected to a compressive load, C-1-1 bears both the interfacial shear load and the peeling load generated by the compressive load moment. However, as the bonding length increases, the moment of the compressive load gradually decreases, and the shear load that C-1-1 can withstand gradually increases. Therefore, when the bonding length increases to a certain extent, C-1-1 does not fail. The failure of C-1-2 and C-1-3 usually occurs after the damage to C-1-1, when the shear component of the compressive load is not on the same plane as the interface of C-1-2 and C-1-3. As a result, C-1-2 and C-1-3 are more prone to peeling. When C-1-2 and C-1-3 failed, Metal (3.00 mm) continued to carry compressive loads, resulting in deformation. When C-1 did not completely fail, the compressive load was transferred to Metal (8.00 mm), which deformed the overall structure.

Figure 17 can only predict failure modes under compressive loading of metal-composite L-joints of different bond lengths before the ultimate load. The metal-composite L-joint designed in the paper has a large residual strength after reaching the ultimate load. In order to be able to clearly analyze the entire failure process of the L-joint, a prediction diagram between compression displacement and failure mode was also made, as shown in Figure 18.

The solid line in Figure 18 represents the compressive displacement connection line between C-1 and the onset of failure of the metal part. It can be seen from Figure 18 that the initial failure displacement of C-1-1 is relatively stable, and the value is small. After C-1-1 was damaged, C-1-2 and C-1-4 suffered shear damage. Overall, their damage initiation displacement was smaller than that of C-1-3. The initial damage of C-1-2, C-1-3, and C-1-4 in Figure 18 are mixed, and this phenomenon is caused by the different locations of stress concentrations due to different bond lengths.

In Figure 18, Metal (3.00 mm) had an initial failure, indicating that C-1-2 and C-1-3 had completely failed. Metal (3.00 mm) undergoes plastic deformation (plastic deformation exceeding 0.01 is the beginning of damage), and the plastic deformation of Metal (3.00 mm) can be divided into three stages, as shown in A, B and C in Figure 18. The initial damage displacement of Metal (3.00 mm) in stage A was small, and the bearing capacity of the metal-composite L-joint was weak. The initial damage displacement of Metal (3.00 mm) in stage B was relatively stable, and the maximum displacement and residual strength of the metal-composite L-joint were higher. In stage C, the initial damage displacement of Metal (3.00 mm) increased with the increase in the bonding length. The damage in this part was the mixed damage of Metal (3.00 mm) and Metal (8.00 mm). Both the maximum displacement and the residual strength of the metal-composite L-joint were improved due to the involvement of Metal (8.00 mm) in the bearing load.

In Figure 18, Metal (8.00 mm) begins to deform plastically, which indicates that C-1-1 has not completely failed. When Metal (8.00 mm) fails, the overall structure of the metal-composite L-joint is bent, and its included angle is less than 90°. When the bonding length is less than 100.00 mm, the main failure mode of the L-joint changes from Metal (8.00 mm), bending to Metal (3.00 mm), due to the failure of C-1-1; when the bonding length is greater than 100.00 mm (as C-1-1 did not fail) the main failure mode of the L-joint would be the continuous bending of Metal (8.00 mm). The initial displacement of the damage was small, which would result in a large bending of the joint.

Based on the above analysis, we can conclude that the bonding length of 100.00 mm is a better choice. It not only has a larger ultimate load and residual strength, but can also undergo a larger compression displacement. However, if the metal-composite L-joint requires no sudden change in load, the bonding length can be designed to be more than 105.00 mm. If the metal-composite L-joint requires freedom from bending deformation of the structural main body, a bonding length between 40.00 and 90.00 mm can be selected.

## 5. Conclusions

In the paper, a metal/composite L-joint with a groove structure was designed and seven groups of specimens with different bonding lengths were fabricated, to study the effect of different bonding lengths on the performance of the joint.
(1)In the simulation analysis of metal/composite L-joints, the groove structure is equivalent to a 0-thickness element layer. The applicability of the simulation method is verified by comparing the ultimate load, displacement and failure mode of the test and simulation.(2)The failure modes of L-joints with different bond lengths are analyzed, and the following conclusions are drawn: When the bonding length is 25.00 mm and 40.00 mm, C-1-1 firstly fails completely, then C-1-2 and C-1-3 fail completely, and most of thestructures of metal components and composite components are de-bonded; when the bonding length is 55.00 mm, 70.00 mm and 85.00 mm, C-1-1 fails first but not completely, while C-1-2 and C-1-3 fail completely, and Metal (3.00 mm) continues to carry the compressive load; when the bonding length is 100.00 mm and 115.00 mm, C-1 does not fail completely. At this time, the metal component bears a large bending load, and the overall structure of the metal/composite L-joint is greatly bent.(3)When the bond length is 25.00–40.00 mm, after the metal/composite L-joint fails, the metal and composite components warp due to excessive compressive displacement. When the bonding length is greater than 100.00 mm, there is no obvious separation of the metal and composite components, and the metal/composite L-joint has obvious overall bending.(4)Finally, the simulation method is used to simulate the compression of more metal/composite L-joints with different bond lengths, and the prediction graphs of failure displacement and failure mode are produced. According to the failure prediction diagram, designers can select the corresponding bonding length according to their own needs; this provides a reference for the design of metal/composite hybrid L-joints.

## Figures and Tables

**Figure 1 polymers-14-01051-f001:**
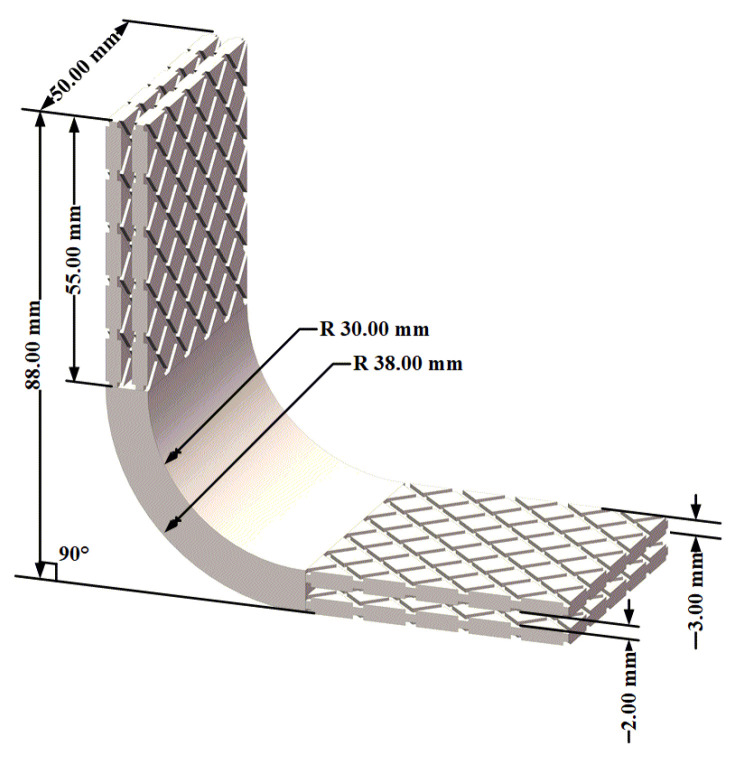
Schematic diagram of L-shaped metal component.

**Figure 2 polymers-14-01051-f002:**
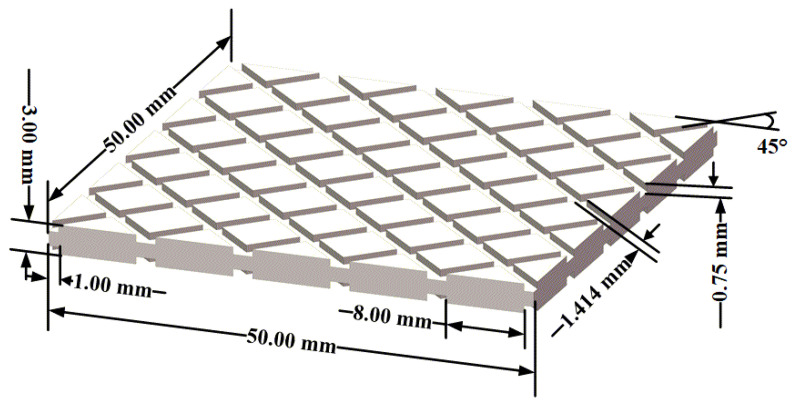
±45° groove structure (area size: 50 mm × 50 mm).

**Figure 3 polymers-14-01051-f003:**
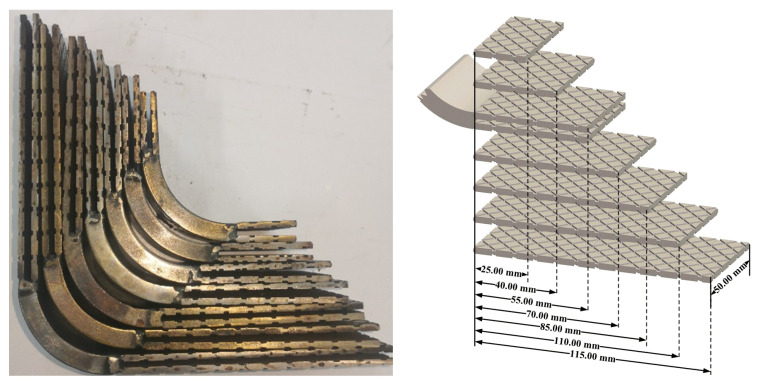
Groove structure with different bonding lengths.

**Figure 4 polymers-14-01051-f004:**
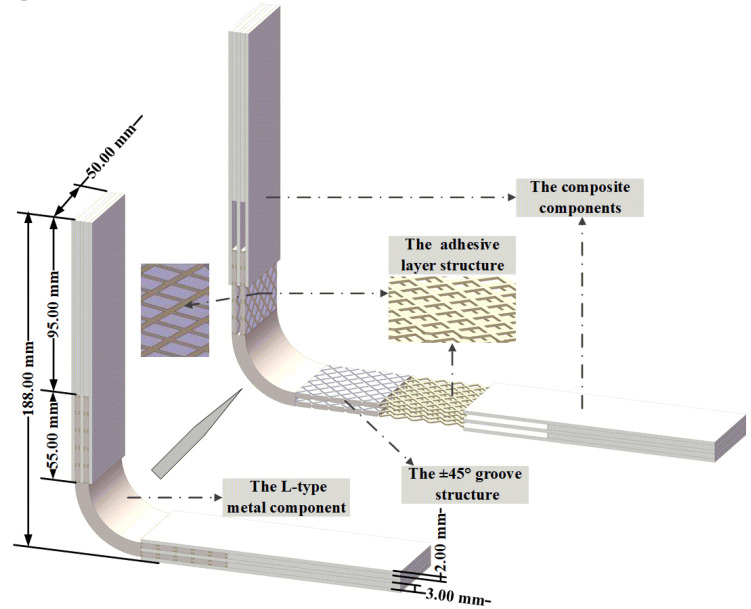
Disassembly schematic diagram of L-joint overall schematic and component.

**Figure 5 polymers-14-01051-f005:**
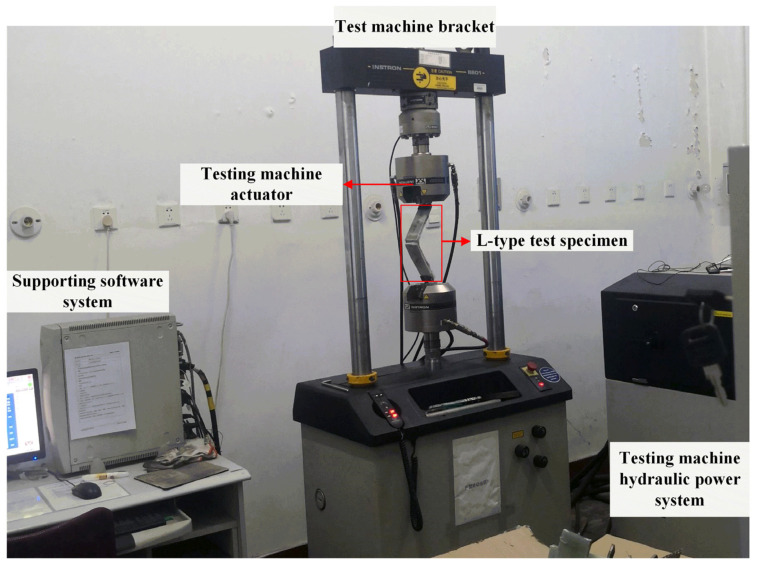
Compression test of metal-composite L-joint and direction of force.

**Figure 6 polymers-14-01051-f006:**
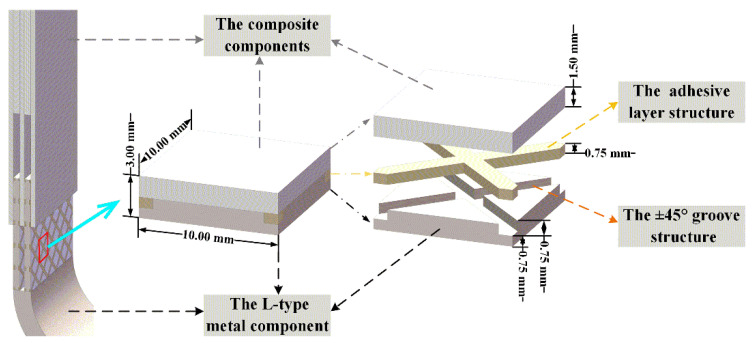
Definition of the RVE.

**Figure 7 polymers-14-01051-f007:**
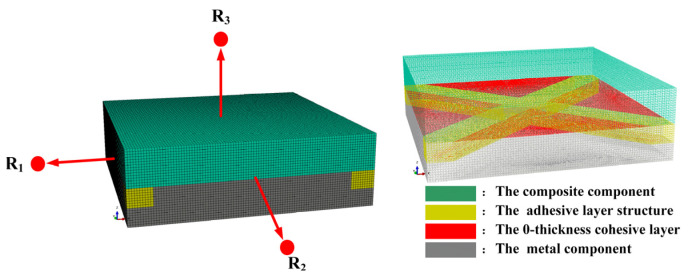
Meshing and components of the RVE.

**Figure 8 polymers-14-01051-f008:**
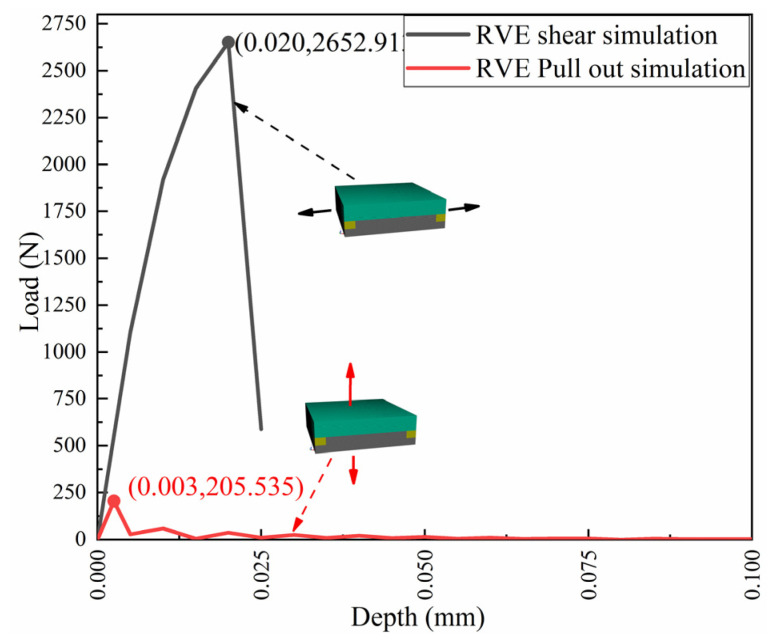
Load–displacement curve of the shear and pull-out tests.

**Figure 9 polymers-14-01051-f009:**
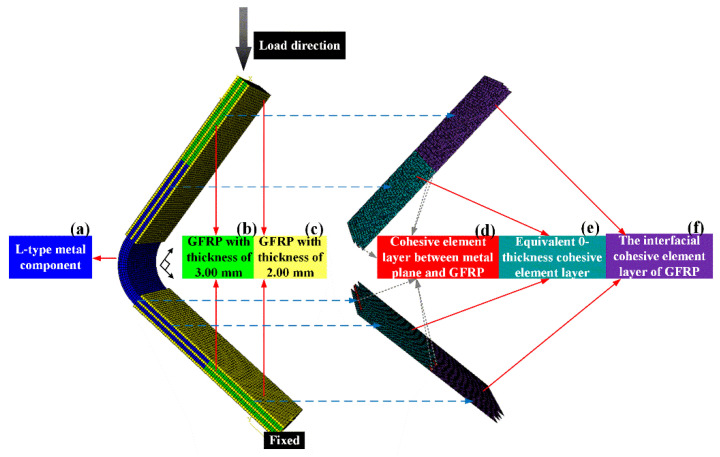
Metal-composite L-joint model mesh and loading conditions: (**a**) L-type metal component; (**b**) GFRP with thickness of 3.00 mm; (**c**) GFRP with thickness of 2.00 mm; (**d**) Cohesive element layer between metal plane and GFRP; (**e**) Equivalent 0-thickness cohesive element layer; (**f**) The interfacial cohesive element layer of GFRP.

**Figure 10 polymers-14-01051-f010:**
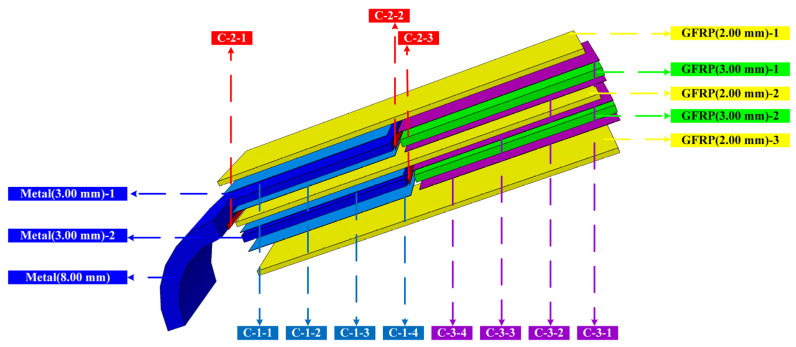
Schematic diagram of disassembly of metal-composite L-joint structure.

**Figure 11 polymers-14-01051-f011:**
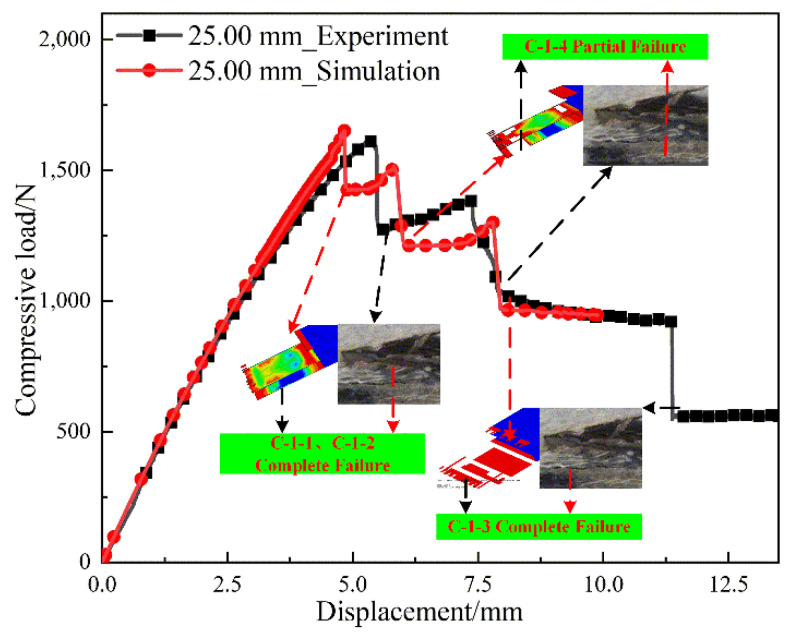
Load–displacement curves for L-joints with a bond length of 25.00 mm.

**Figure 12 polymers-14-01051-f012:**
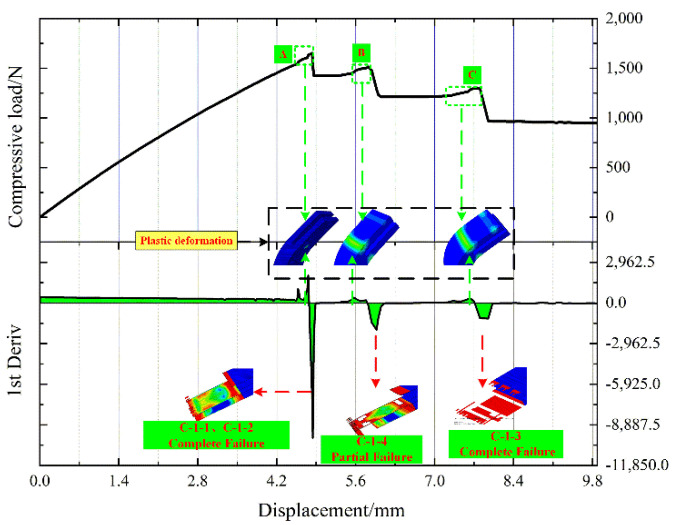
The load–displacement curve and its first derivative curve of the compression simulation of the L-joint, and (**A**) The first load stabilization stage; (**B**) The second load stabilization stage; (**C**) The third load stabilization stage.

**Figure 13 polymers-14-01051-f013:**
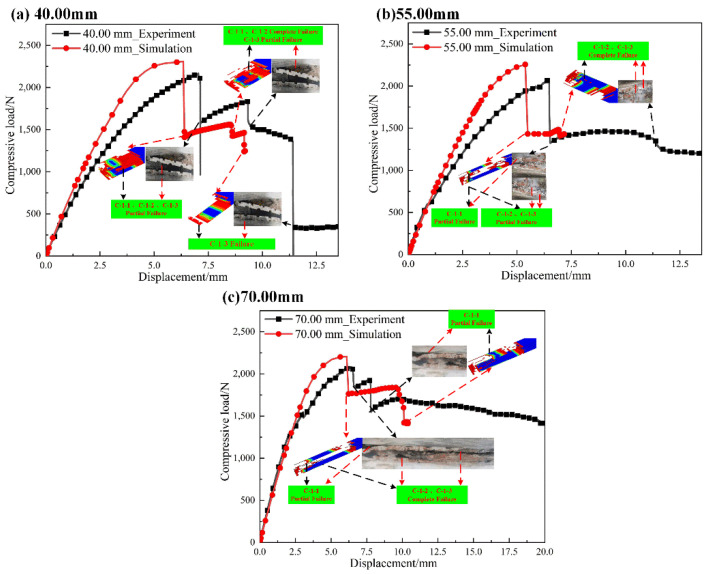
Load–displacement curves for L-joints with different bonding lengths: (**a**) 40.00 mm, (**b**) 55.00 mm and (**c**) 70.00 mm.

**Figure 14 polymers-14-01051-f014:**
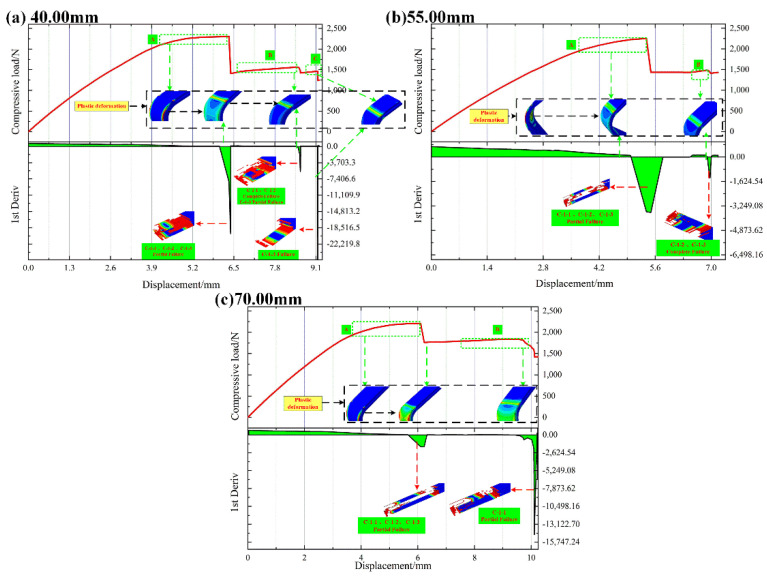
The load–displacement curve and its first derivative curve of the compression simulation of the L-joint with different bonding lengths: (**a**) 40.00 mm, (**b**) 55.00 mm and (**c**) 70.00 mm and (A) The first load stabilization stage; (B) The second load stabilization stage; (C) The third load stabilization stage.

**Figure 15 polymers-14-01051-f015:**
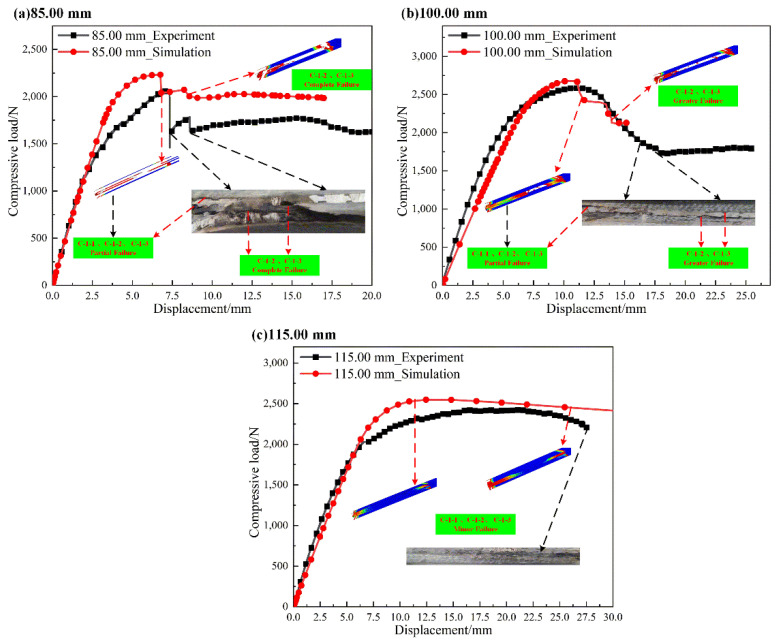
Load–displacement curves for L-joints with different bonding lengths: (**a**) 85.00 mm, (**b**) 100.00 mm and (**c**) 115.00 mm.

**Figure 16 polymers-14-01051-f016:**
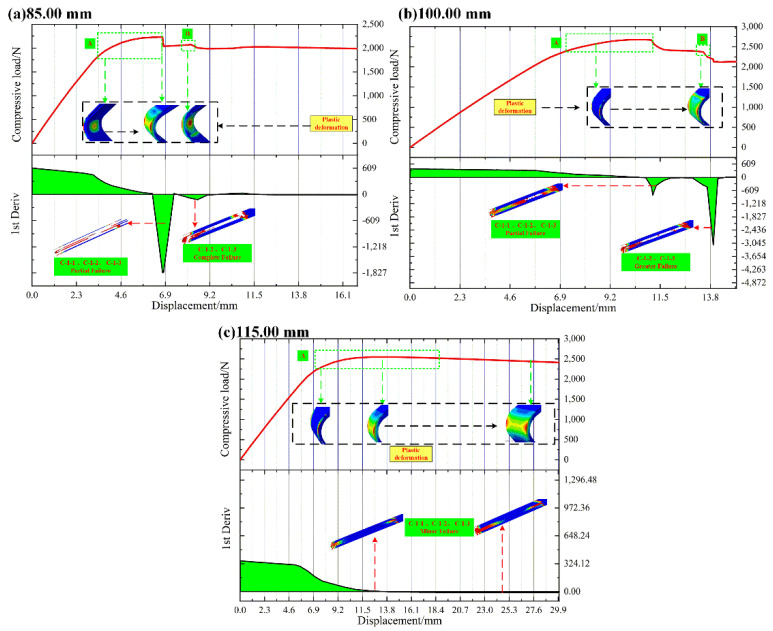
The load–displacement curve and its first derivative curve of the compression simulation of the L-joint with different bonding lengths: (**a**) 85.00 mm, (**b**) 100.00 mm and (**c**) 115.00 mm, and (A) The first load stabilization stage; (B) The second load stabilization stage.

**Figure 17 polymers-14-01051-f017:**
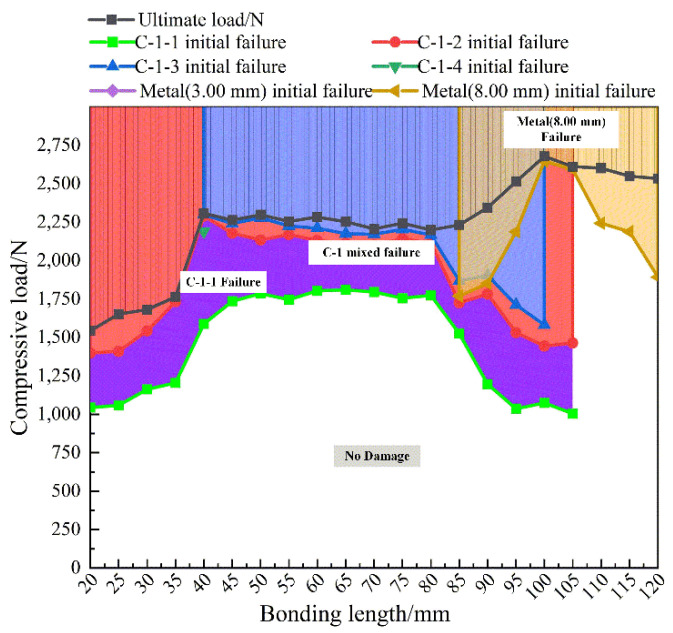
Damage and failure prediction diagram for metal-composite L-joints.

**Figure 18 polymers-14-01051-f018:**
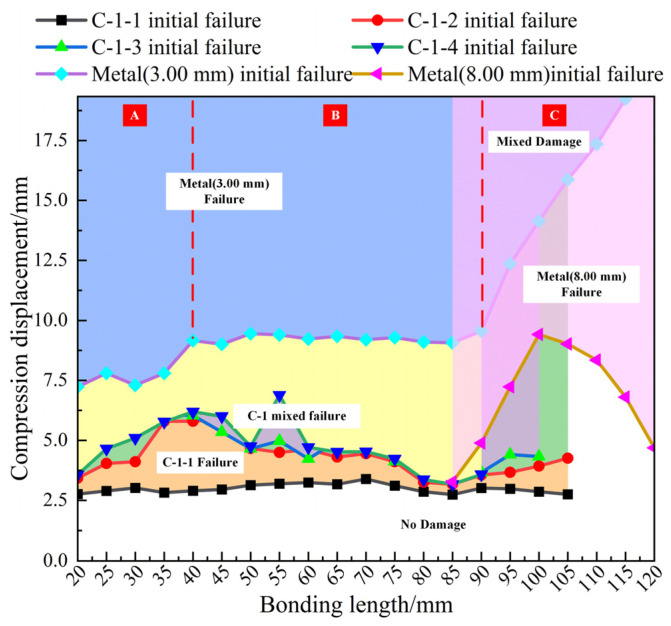
Damage and failure prediction diagram for metal-composite L-joints and Metal (3.00 mm) initial failure displacement fluctuating increasing stage (**A**), stable stage (**B**) and stable increasing stage (**C**).

**Table 1 polymers-14-01051-t001:** Initial stiffness of the interface, damage initiation and propagation parameters.

Initial Stiffness (MPa/mm)			Interlaminar Strength (MPa)			Fracture Toughness (J/mm^2^)		
Kn	Kt	Ks	N	T	S	*G_IC_*	*G_IIC_*	*G_IIIC_*
5811	5811	5811	2.05	26.53	26.53	0.085	2.65	2.65

**Table 2 polymers-14-01051-t002:** Degradation rules for the material properties used in the study.

Failure Mode	Failure Criterion	Material Degradation Criterion
Fiber tensile failure	$\sigma_{11}\geq 0$	$\begin{array}{l} E{’}_{11}=0.07E_{11};\;G{’}_{12}=0.07G_{12};\;G{’}_{13}=0.07G_{13}\\ \;\;\;\;\;\;\;\;\nu {’}_{12}=0.07\nu_{12};\;\nu {’}_{13}=0.07\nu_{13} \end{array}$
Fiber compression failure	$\sigma_{11}<0$	$\begin{array}{l} E{’}_{11}=0.07E_{11};\;G{’}_{12}=0.07G_{12};\;G{’}_{13}=0.07G_{13}\\ \;\;\;\;\;\;\;\;\nu {’}_{12}=0.07\nu_{12};\;\nu {’}_{13}=0.07\nu_{13} \end{array}$
Matrix tensile failure	$\sigma_{22}+\sigma_{33}\geq 0$	$\begin{array}{l} E{’}_{22}=0.2E_{22};G{’}_{12}=0.2G_{12};G{’}_{23}=0.2G_{23}\\ \;\;\;\;\;\;\;\;\;\;\;\;\;\nu {’}_{12}=0.2\nu_{12};\nu {’}_{23}=0.2\nu_{23} \end{array}$
Matrix compression failure	$\sigma_{22}+\sigma_{33}<0$	$\begin{array}{l} E{’}_{22}=0.4E_{22};G{’}_{12}=0.4G_{12};G{’}_{23}=0.4G_{23}\\ \;\;\;\;\;\;\;\;\;\;\;\;\;\nu {’}_{12}=0.4\nu_{12};\nu {’}_{23}=0.4\nu_{23} \end{array}$
Tensile delamination failure	$\sigma_{33}\geq 0$	$\begin{array}{l} E{’}_{33}=0.2E_{33};G{’}_{13}=0.2G_{13};G{’}_{23}=0.2G_{23}\\ \;\;\;\;\;\;\;\;\;\;\;\;\nu {’}_{13}=0.2\nu_{13};\nu {’}_{23}=0.2\nu_{23} \end{array}$
Compression delamination failure	$\sigma_{33}<0$	$\begin{array}{l} E{’}_{33}=0.2E_{33};G{’}_{13}=0.2G_{13};G{’}_{23}=0.2G_{23}\\ \;\;\;\;\;\;\;\;\;\;\;\;\nu {’}_{13}=0.2\nu_{13};\nu {’}_{23}=0.2\nu_{23} \end{array}$

**Table 3 polymers-14-01051-t003:** The comparison of the ultimate loads and corresponding displacements of different L-joint experiments and simulation curves.

Bonding Length	Ultimate Load/N			Displacement Corresponding to Ultimate Load/mm		
	Experiment	Simulation	Deviation Ratio/%	Experiment	Simulation	Deviation Ratio/%
40.00 mm	2166.25	2308.64	6.57	6.95	6.34	8.78
55.00 mm	2075.22	2255.98	8.71	6.41	5.38	16.06
70.00 mm	2071.91	2206.83	6.51	6.42	6.10	4.95

**Table 4 polymers-14-01051-t004:** The comparison of the ultimate loads and corresponding displacements of different L-joint experiments and simulation curves.

Bonding Length	Ultimate Load/N			Displacement Corresponding to Ultimate Load/mm		
	Experiment	Simulation	Deviation Ratio/%	Experiment	Simulation	Deviation Ratio/%
85.00 mm	2050.47	2232.69	9.89	7.35	6.76	8.03
100.00 mm	2615.07	2677.64	2.39	11.56	10.83	6.31
115.00 mm	2422.13	2547.18	5.16	16.57	14.82	10.56

**Table 5 polymers-14-01051-t005:** Simulated failure mode versus compressive load.

Failure Structure	C-1-1	C-1-2	C-1-3	C-1-4	Metal (8.00 mm)	Ultimate Load
Bonding length 25.00 mm	1058.38 N	1409.94 N				1652.02 N
Bonding length 40.00 mm	1588.23 N	2290.35 N	2300.11 N	2190.23 N		2308.61 N
Bonding length 55.00 mm	1746.51 N	2169.06 N	2224.84 N			2255.95 N
Bonding length 70.00 mm	1796.48 N	2101.12 N	2172.15 N			2206.80 N
Bonding length 85.00 mm	1525.68 N	1725.89 N	1868.06 N		1770.53 N	2232.63 N
Bonding length 100.00 mm	1075.14 N	1444.15 N	1581.67 N		2647.89 N	2679.46 N
Bonding length 115.00 mm					2188.98 N	2548.96 N

**Table 6 polymers-14-01051-t006:** Simulated failure mode versus displacement.

Failure Structure	C-1-1	C-1-2	C-1-3	C-1-4	Metal (3.00 mm)	Metal (8.00 mm)
Bonding length 25.00 mm	2.89 mm	4.04 mm	8.03 mm	6.12 mm	8.03 mm	
	4.48 mm	4.48 mm				
Bonding length 40.00 mm	6.40 mm	6.40 mm	6.40 mm	6.40 mm	9.16 mm	
	8.61 mm	8.61 mm	9.16 mm			
Bonding length 55.00 mm	5.79 mm	5.79 mm	5.79 mm		7.00 mm	
		7.00 mm	7.00 mm			
Bonding length 70.00 mm	6.32 mm	6.32 mm	6.32 mm		6.32 mm	
	10.21 mm					
Bonding length 85.00 mm	7.35 mm	7.35 mm	7.35 mm		9.07 mm	
		9.07 mm	9.07 mm			
Bonding length 100.00 mm	12.00 mm	12.00 mm	12.00 mm			14.14 mm
Bonding length 115.00 mm						12.45 mm

## Data Availability

The data presented in this study are available on request from the corresponding author.

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
