# Peer review of "Failure-Mode Shift of Metal/Composite L-Joint with Grooved Structure under Compressive Load"

_polymers, 2022, doi:10.3390/polym14051051_

Round 1
Reviewer 1 Report
The author has discussed about failure mode of metal-composite joint under compressive load.
Thsi manuscript can be classified as an well-made research paper, and also contains high-degree of the practical scientific contribution in this research fields, especially damage & failure prediction diagram for joint which the athuor has proposed.
Author Response
Thank you very much for your review and recognition.

Reviewer 2 Report
In this paper, failure analysis of L-shape metal/composite is studied. The structure is under the compressive load. This paper has a good topic but need to below revisions:
1-The failure analysis is studied but the damage factor for the structure has not been reported. This factor using a theory such as Hook-Brown or other models should be added to the revised paper.
2-The boundary conditions of L-shape structure should be clarified and its effect should be studied on the results.
3-The introduction section has not a good relation between sub-sections. hence, should be modified.
4-The interaction of metal and composite is essential to study.
5-From the physical side, the results need to modification.
6-Some abbriviations should be defined such as VARI and etc
Author Response
The authors have carefully revised following your comments and suggestions. Since there are charts and formulas, the details are in the attachment.

Round 2
Reviewer 2 Report
This paper can be accepted for publication